# The Class I HDAC Inhibitor Valproic Acid Strongly Potentiates Gemcitabine Efficacy in Pancreatic Cancer by Immune System Activation

**DOI:** 10.3390/biomedicines10030517

**Published:** 2022-02-22

**Authors:** Amber Blaauboer, Peter M. van Koetsveld, Dana A. M. Mustafa, Jasper Dumas, Fadime Dogan, Suzanne van Zwienen, Casper H. J. van Eijck, Leo J. Hofland

**Affiliations:** 1Department of Surgery, Erasmus Medical Center, 3015 GD Rotterdam, The Netherlands; a.blaauboer@erasmusmc.nl (A.B.); c.vaneijck@erasmusmc.nl (C.H.J.v.E.); 2Department of Internal Medicine, Division of Endocrinology, Erasmus Medical Center, 3015 GD Rotterdam, The Netherlands; p.vankoetsveld@erasmusmc.nl (P.M.v.K.); f.dogan@erasmusmc.nl (F.D.); svzwienen@hotmail.com (S.v.Z.); 3Tumor Immuno-Pathology Laboratory, Department of Pathology, Erasmus Medical Center, 3015 GD Rotterdam, The Netherlands; d.mustafa@erasmusmc.nl (D.A.M.M.); j.dumas@erasmusmc.nl (J.D.)

**Keywords:** pancreatic cancer, gemcitabine resistance, valproic acid, immune activation

## Abstract

Background: Gemcitabine efficacy in pancreatic cancer is often impaired due to limited intracellular uptake and metabolic activation. Epi-drugs target gene expression patterns and represent a promising approach to reverse chemoresistance. In this study, we investigate the chemosensitizing effect of different epi-drugs when combined with gemcitabine in pancreatic cancer. Methods: Mouse KPC3 cells were used for all experiments. Five different epi-drugs were selected for combination therapy: 5-aza-2′-deoxycytidine, hydralazine, mocetinostat, panobinostat, and valproic acid (VPA). Treatment effects were determined by cell proliferation and colony forming assays. Expression of genes were assessed by real-time quantitative PCR. The most promising epi-drug for combination therapy was studied in immune competent mice. Intratumor changes were defined using NanoString PanCancer panel IO360. Results: All epi-drugs, except hydralazine, potentiated the gemcitabine response in KPC3 cells (range decrease IC_50_ value 1.7–2-fold; *p* < 0.001). On colony formation, the cytotoxic effect of 0.5 ng/mL gemcitabine was 1.4 to 6.3 times stronger (*p* < 0.01). Two out of three drug-transporter genes were strongly upregulated following epi-drug treatment (a range fold increase of 17–124 and 9–60 for *Slc28a1* and *Slc28a3*, respectively; all *p* < 0.001). VPA combined with gemcitabine significantly reduced tumor size with 74% compared to vehicle-treated mice and upregulated expression of immune-related pathways (range pathway score 0.86–1.3). Conclusions: These results provide a strong rationale for combining gemcitabine with VPA treatment. For the first time, we present intratumor changes and show activation of the immune system. Clinical trials are warranted to assess efficacy and safety of this novel combination in pancreatic cancer patients.

## 1. Introduction

Pancreatic cancer represents the fourth most lethal cancer in the Western world, with a 5-year survival rate of 8.5% [1]. The incidence rate of pancreatic cancer is slowly rising and, as a consequence, it is expected to be the second leading cause of cancer-related death by 2030 [2]. Upfront surgery is currently the only curative-intent treatment option. However, less than 20% of the patients present with resectable disease, and relapse rates after surgery are high [3,4].

To increase survival rates following surgery, gemcitabine (2′,2′-difluoro 2′deoxycytidine, dFdC) has long been the standard first-line chemotherapy for patients with resectable disease [5]. However, its therapeutic value is substantially limited due to treatment resistance [6]. Despite the introduction of other more effective chemotherapeutic regimens, such as FOLFIRINO.X.; gemcitabine/nab-paclitaxel and gemcitabine/capecitabine, gemcitabine alone is still recommended for (elderly) patients with comorbidities, as it is much better tolerated with less toxicity compared to the other chemotherapeutic agents [7,8].

Gemcitabine is a nucleoside analogue with various inhibitory actions on DNA synthesis as it can be incorporated into the DNA, resulting in masked chain termination and apoptosis [9]. It possesses distinct characteristics in terms of metabolism and pharmacokinetics. To be explicit, there are two main classes of genes that are crucial for gemcitabine’s antitumor actions. First, the membrane transporter-coding genes, whose products are responsible for intracellular uptake, and second, the drug metabolism-coding genes, which catalyze the activation or inactivation of gemcitabine. These genes are typically downregulated in pancreatic cancer, resulting in reduced effectiveness of gemcitabine [10,11,12,13].

Recently, epi-drugs have emerged as a novel and promising approach to reactivate gene expression [14]. Epi-drugs are chemical compounds that target specific enzymes, which are necessary for the maintenance and establishment of epigenetic modifications, with the main strategy being the inhibition of DNA methyltransferases (DNMTs) and histone deacetylates (HDACs). As opposed to modifications of the genomic sequence itself, epigenetic modifications are reversible, since they do not modify the base pair sequence of the DNA, but modify the DNA conformation and, consequently, modify gene expression patterns [14]. The biochemical reversibility of these modifications has led to attempts at therapeutic application that primarily target these mechanisms via inhibition of DNMTs (DNMTi) and HDACs (HDACi).

For many decades, pancreatic cancer was believed to develop through the sequential accumulation of genetic mutations. However, the identification of epigenetics has changed this dominant paradigm, as they play a significant role in carcinogenesis as well as in the response of tumor cells to chemotherapy [15]. In fact, epigenetic modifications are considered as far more prevalent in cancer than genetic mutations and, most importantly, are reversible, lending themselves as potential therapeutic targets. Consistently, DNMT and HDAC activity is higher in pancreatic cancer compared to normal pancreas, and their importance increases even more upon tumor progression [16,17,18,19].

Here, we assessed the potential ability of different DNMTis and HDACis to improve gemcitabine response, analyzed the effect on the expression of gemcitabine transporter and metabolizing genes, and subsequently validated the most promising epi-drug for combination therapy with gemcitabine in a representative murine model of human pancreatic cancer.

## 2. Materials and Methods

### 2.1. Cell Lines and Compounds

The mouse pancreatic cancer KPC3 cell line, derived from a primary tumor of a female KrasG12D/+;Trp53R172H/+;Pdx-1-Cre (KPC) mouse, was kindly provided by dr. van Montfoort (Department of Medical Oncology, Leiden University Medical Center, the Netherlands). Cells were authenticated by short-tandem-repeat DNA profiling and confirmed as mycoplasma-free. Culture conditions were described in detail previously [20]. In short, cells were routinely incubated with humidified incubation at 37 °C and 5% CO_2_ and cultured in RPMI 1640 supplemented with 10% fetal calf serum (FCS), penicillin (1 × 10^5^ U/L), and L-glutamine (2 mmol/L). KPC3 cells were plated in 24-wel plate at the appropriate density to obtain 80% confluency at the end of the experiment. The next day, cells were treated with the various compounds and incubated for seven days. The medium and compounds were refreshed after three days. Cell culture experiments were carried out at least twice in quadruplicate. 5-aza-2′-deoxycytidine (5-AZA), gemcitabine, hydralazine, and valproic acid (VPA) stock dilutions were diluted in distilled water. Mocetinostat and panobinostat were diluted in 40% dimethylsulfoxide. All compounds were obtained from Sigma-Aldrich (Zwijndrecht, The Netherlands). Control cells were vehicle-treated (distilled water or DMSO-final concentration 0.4%).

### 2.2. Cell Proliferation Assay

Dose-response curves on cell growth were obtained to assess IC_25_ and IC_50_ inhibitory concentrations of the indicated epi-drugs for combination experiments with gemcitabine. Dose-dependent effects of gemcitabine, with or without epi-drugs, were assessed on cell amounts after seven days of treatment. After treatment, the medium was removed and cells were collected for total DNA measurement (as a measure of cell number), performed by the bisbenzimide fluorescent dye (Hoechst 33258, Boehring Diagnostics, La Jolla, CA, USA) as previously described [21].

### 2.3. Colony-Forming Assay

Plates were coated with 1 mL poly-l-lysine (10 μg/mL), where after 500 KPC-3 cells were plated in a six wells plate. After one day, drug treatment was initiated. Media were removed and refreshed without drugs after seven days. After two weeks, cells were washed and stained with haematoxylin. Number and size of colonies were measured using a MultiImage light cabinet (Alpha Innotech, San Leandro, CA, USA) and the ImageJ software. Plating efficiency (PE) was calculated as the mean number of colonies/number of plated cells for control cultures not exposed to drugs. The surviving fraction was calculated as the mean number of colonies/(number of inoculated cells × PE). The effect on the surviving fraction and colony size represent the cytotoxic and cytostatic effects, respectively.

### 2.4. Real-Time Quantitative PCR

RNA isolation, cDNA synthesis, and RT-qPCR were performed as previously described, only with other primers (Appendix A) [22]. Two housekeeping genes were used to normalize *mRNA* levels using the Vandesompele method: *hypoxanthine-guanine phosphoribosyl transferase 1* (*Hprt1*) and *beta glucuronidase* (*Gusb*) (Thermo Fisher Scientific, Breda, the Netherlands) [23]. We calculated PCR efficiencies (E) for the primer-probe combinations (Appendix A) and calculated relative expression of genes using the comparative CT method 2^−ΔΔCt^. To calculate fold-changes for initial undetectable genes, CT values were set at 40.

### 2.5. Mice

The 8–10 weeks aged male C57BL/6 mice were purchased from Charles River Laboratories. All mice were housed in groups of seven. All mouse experiments were controlled by the animal welfare committee (IvD) of the Erasmus University Medical Center (Rotterdam) and approved by the national central committee of animal experiments (CCD) under the permit number AVD101002017867, in accordance with the Dutch Act on Ani-mal Experimentation and European Union (EU) Directive 2010/63/EU.

### 2.6. In Vivo Experiments

Mice were randomized in four groups and subcutaneously injected in the flank with 100.000 KPC3 cells (passage number 3) in 100 µL PBS/0.1% BSA. Cultured KPC3 cells were harvested at 80% confluency and only single-cell suspensions of greater than 90% viability were used for injection. Tumor size and body weight were measured twice weekly. Tumor volume was calculated as (width^2^ × length)/2 using a caliper. Treatment was started when tumor volumes reached ~50 mm^3^. Mice in the control group and in the VPA monotherapy group received an intraperitoneal (i.p.) injection of 100 μL of distilled water or 500 mg/kg VPA daily. Mice randomized to the gemcitabine monotherapy group received an i.p. injection of 50 mg/kg gemcitabine two times a week (days two and five). Mice in the combination group received a daily an injection of 500 mg/kg VPA i.p. upon start of the treatment, and on days two and five they received an i.p. injection of 50 mg/kg gemcitabine.

### 2.7. Necropsy Procedures

Mice were sacrificed by cervical dislocation under isoflurane anesthesia when tumor volume reached 1000 mm^3^ or when the wellbeing of the mice could no longer be maintained. During necropsy, tumors were resected and tumor volumes were measured. Tumors were divided into two parts. One part was snap-frozen in liquid nitrogen and one part fixed in freshly prepared 4% formaldehyde solution and prepared for paraffin sectioning (FFPE).

### 2.8. NanoString Analysis

RNA was extracted from tumor tissues using the RNeasy Plus Micro kit (Qiagen, Venlo, The Netherlands) according to the manufacturer’s instructions. RNA samples were eluted in RNA free water and stored at −80 °C until further measurements. RNA Quality Control (QC) was measured using the 2100 Bioanalyzer (Agilent, Santa Clara, CA, USA). The total RNA concentration was corrected to include fragments seized between 300 and 4000 nucleotides. A total of 200 ng RNA was hybridized to the PanCancer IO 360 Panel (NanoString Technology, Seattle, WA, USA) for 17 h at 67 °C. Data analysis was performed using the advanced analysis module (version 2.0) of nSolver™ software (version 4.0, NanoString Technology). Based on expression stability and minimum variance, eight housekeeping genes (out of 11) were selected for normalization with the geNorm algorithm embedded in the advanced analysis module (Appendix A). The threshold of expression was calculated as twice the average expression of the negative controls. Genes that showed an expression count below the threshold in >80% of the samples were excluded from further analysis. The normalized data were log^2^ transformed and the differentially expressed genes were identified using simplified negative binomial models, mixture negative binomial models, or log-linear models based on the convergence of each gene. The adjusted *p*-value was calculated using the Benjamini-Hoghberg method. Genes were considered differentially expressed when the adjusted *p*-value < 0.05.

### 2.9. Statistical Analysis

GraphPad Prism version 3.0 (GraphPad Software, San Diego, CA, USA) was used for statistical analysis. One-way ANOVA test with Tukey’s multiple comparisons test was used for comparisons among treatment groups. Regarding in vivo experiments, differences between groups were evaluated by the Mann-Whitney *t*-test. In all analyses, values of *p* < 0.05 were considered as significant. Data are indicated as mean ± SEM, unless specified otherwise.

## 3. Results

### 3.1. Effect of Epi-Drug Monotherapy In Vitro

All epi-drugs induced a dose-dependent inhibitory response. 5-AZA inhibited cell proliferation at much lower concentrations (IC_50_ 0.078 µM; 95% confidence interval [CI] 0.071–0.087) than hydralazine (IC_50_ 34.17 µM; 95% CI 32.8–35.6; *p* < 0.001). IC_50_ values of HDACis were lower for panobinostat (0.11 µM; 95% CI 0.097–0.12) compared to mocetinostat (353 µM; 95% CI 317–393; *p* < 0.001) and VPA (1098 µM; 95% CI 1032–1169; *p* < 0.001) (Figure 1).

### 3.2. Effect of Epi-Drugs on Gemcitabine Sensitivity In Vitro

#### 3.2.1. Cell Proliferation Assay

Next, we examined whether these epi-drugs could sensitize KPC3 cells for gemcitabine treatment, as reflected by a decrease in IC_50_ value. IC_25_ and IC_50_ values were chosen for combination experiments with gemcitabine (Appendix A). All epi-drugs, except hydralazine, increased the response to gemcitabine (Figure 2, Appendix A). This increase was already evident when IC_25_ values were used (range decrease in IC_50_ value 1.3–1.8-fold; all *p* < 0.001 vs. untreated control cells). The addition of IC_50_ values did not further enhance the sensitivity of KPC3 cells for gemcitabine treatment compared to IC_25_-treated cells (range decrease in IC_50_ value 1.7–2-fold; all *p* < 0.001 vs. untreated control cells) (Figure 2, Table 1).

VPA induced the strongest sensitizing effect when added to different doses of gemcitabine. To be explicit, the inhibitory effect of 0.5 ng/mL and 1 ng/mL gemcitabine was increased by 44% and 23%, respectively, when co-treated with IC_50_ VPA (*p* < 0.001 vs. untreated control cells) (Appendix A, lower right panel).

#### 3.2.2. Colony-Forming Assay

The chemosensitizing effect of epi-drugs appeared to be primarily cytotoxic, as stated by an effect on the number of colonies rather than an effect on colony size (Figure 3 and Appendix A). To avoid strong cytotoxicity, low concentrations of gemcitabine (0.5 ng/mL and 1 ng/mL) were used for combination experiments (Figure 3). The effect of epi-drug monotherapy (IC_25_ and IC_50_) are presented in Appendix A. All epi-drugs, except hydralazine, potentiated the cytotoxic effect of gemcitabine in KPC3 cells. While 0.5 ng/mL gemcitabine had no significant inhibitory effect on colony number, its cytotoxic effect was strongly enhanced when co-treated with epi-drugs (range increase 1.6–2.3-fold and 1.4–6.3-fold in IC_25_ and IC_50_ epi-drug treated cells, respectively; *p* < 0.01 vs. un-treated control cells). The cytotoxic effect of 1 ng/mL gemcitabine, which inhibited colony formation with 50%, was approximately three times stronger when co-treated with epi-drugs (*p* < 0.001). The cytostatic effect of gemcitabine was only slightly enhanced by VPA co-treatment (an approximately 1.5-fold increase; *p* < 0.05 vs. untreated control cells) (Appendix A).

### 3.3. Effect of epi-Drugs on the Expression of Genes Involving Gemcitabine Uptake and Metabolism

As a potential mechanism underlying the chemosensitizing effect, we hypothesized that epi-drugs in-crease expression of genes involving intracellular uptake and metabolic activation of gemcitabine. Our hypothesis was investigated by measuring drug transporter genes. The expression of *Slc29a1* was low in KPC3 cells and even undetectable for *Slc28a1* and *Slc28a3* (Appendix A). All epi-drugs, except hydralazine, increased *mRNA* expression of both *Slc28a1* and *Slc28a3* (range fold increase 17–124 and 9–60 for *Slc28a1* and *Slc28a3* respectively; all *p* < 0.01) (Figure 4). Expression of *Slc29a1* was only modestly upregulated by mocetinostat (3-fold increase; *p* < 0.001) and panobinostat (1.3-fold increase, *p* < 0.01).

Relative baseline *mRNA* expression of the activating and inactivating genes ranged from 0.2–11 and was not extensively altered by epi-drugs. Expression of the inactivating *Cda* gene was upregulated by all three HDACis, with the strongest upregulation after IC_50_ mocetinostat treatment (7-fold, *p* < 0.001) (Figure 4). Contrary to this, 5-AZA and hydralazine slightly downregulated expression of *Cda* (both had a 0.3-fold decrease; *p* < 0.01).

### 3.4. In Vivo Validation of VPA Combined with Gemcitabine in Immune-Competent Mice

Based on the in vitro results, VPA was chosen as the most promising epi-drug to potentiate the antitumor response of gemcitabine in a subcutaneous KPC3 pancreatic cancer model. C57BL/6 mice were randomized into four treatment arms: vehicle (H_2_O), daily 500 mg/kg VPA, twice-weekly 50 mg/kg gemcitabine, or the combination of daily 500 mg/kg VPA plus twice-weekly 50 mg/kg gemcitabine (Figure 5A).

Combination therapy resulted in significant tumor growth control over time and smaller tumor volumes when compared with untreated mice, while monotherapy with VPA or gemcitabine did not. After 21 days of treatment, tumor volume was reduced by 74% compared to vehicle-treated mice (1239 vs. 324 mm^3^; *p* = 0.003) (Figure 5B–D). In the end, total body weight was reduced by 7% in combination-treated mice (*p* < 0.05 compared to control) (Figure 5E).

Expression of transporter and metabolizing genes of gemcitabine were very low in untreated KPC3 tumors and were not affected by any treatment (Appendix A).

### 3.5. PanCancer IO 360 Gene Expression Panel

To gain more insight into the intratumor changes, we performed a targeted gene expression array on tumor samples of treated and untreated mice.

The overview of the differentially expressed genes upon treatment compared to untreated mice is summarized in Figure 6A. In total, the expression of 40 and 36 genes was significantly altered by gemcitabine and combination therapy, respectively, with an overlap of 15 genes between both treatment groups (Figure 6B, Appendix A). Although VPA monotherapy induced no significant changes, a similar trend in gene expression was observed compared to combination-treated tumors (Appendix A).

The primary difference between gemcitabine-treated mice and combination-treated mice was found in the expression of the immune-related genes. All included immune pathways were upregulated by combination therapy, whereas gemcitabine monotherapy induced a suppressive effect (Figure 6C). To be explicit, immune pathway scores ranged between 0.86 and 1.3 in the combination group and between −0.46 and −1.22 in the gemcitabine group compared to untreated mice (Figure 6D).

In total, four immune-related genes were upregulated by gemcitabine (*S1008a*, *Lilra5*, *Ctsw*, and *Tbx21*) and seven genes were downregulated (*Nos2*, *Cdkn1a*, *Igf2r*, *H2-k1*, *Pvr*, *Irf2i*, and *Mtor*) (Appendix A). In the combination group, nine genes (*Lilra5*, *Clec7a*, *H2-k1*, *Itgal*, *Tbx21*, *Il18bp*, *Stat2*, *Irf3*, and *Nfkb2*) were upregulated and four genes (*Tnfrsf11b*, *Ccnd2*, *Prkca*, and *Il22ra1*) were downregulated. In both groups, *Lilra5* (log 2-fold change 1.43 and 2.02 respectively) and *Tbx21* (log 2-fold change 1.06 and 0.947 respectively) were upregulated (all *p* < 0.05 compared to control). Contrary, *H2-k1* was downregulated by gemcitabine (log 2-fold change −0.581; *p* = 0.0427 compared to control), but upregulated in the combination group (log 2 fold change 1.31; *p* < 0.00436 compared to control).

Importantly, pro-tumoral pathways, such as angiogenesis and metastasis, were downregulated in all three treatment arms compared to untreated tumors and did not differ from each other (Figure 6C).

## 4. Discussion

Despite the increased use of more effective chemotherapeutic combinations, such as FOLFIRINOX or gemcitabine with either nab-paclitaxel or capecitabine, resistance is still a major impediment to the successful treatment of pancreatic cancer [7]. Thereby, these new treatment regimens are accomplished with severe side effects and often grade 3–4 infections [8]. Gemcitabine alone is still being recommended as first-line adjuvant chemotherapy for (elderly) patients with comorbidities, as it is much better tolerated with less toxicities [7]. However, there is an urgent need to improve gemcitabine efficacy, while preventing overtreatment and unnecessary adverse side-effects.

Epi-drugs have emerged as a promising therapeutic approach to reactivate epigenetically silenced genes that are involved in drug resistance mechanisms [14]. Hence, we studied the potential chemosensitizing effect of different epi-drugs combined with gemcitabine in the KPC3 pancreatic cancer mouse model. A low concentration VPA, which is a well-tolerated class I HDACi, strongly potentiated the antitumor effect of gemcitabine in vitro and in vivo. The tumor size of immune competent mice was evidently smaller and showed increased expression of numerous immune-related pathways.

Previously, the antitumor effects of epi-drugs as a single drug were extensively demonstrated in various cancer models, including pancreatic cancer [24,25,26]. Numerous biological processes are controlled by epi-drugs e.g., cell differentiation, cell cycle arrest, apoptosis as well as energy metabolism. Here, we studied the effect two DNMTis (5-AZA and hydralazine) and three HDACis (mocetinostat, panobinostat, and VPA) on the proliferation of KPC3 cells. All epi-drugs effectively inhibited tumor cell growth in a dose-dependent manner, all with IC_50_ values within the range of achievable plasma concentrations after administration in humans.

KPC3 cells were most sensitive for 5-AZA treatment (IC_50_ 0.078 µM), which is a well-known chemo-therapeutic agent and approved by the FDA for the treatment of several cancers, e.g., acute myeloid leukemia and myelodysplastic syndrome [27]. The direct antiproliferative effects of 5-AZA can be explained by two mechanisms [28]. At low doses, 5-AZA induces DNA hypomethylation via DNMT1 inhibition, causing reactivation of silenced tumor-suppressive genes. At high doses, 5-AZA prevents DNA synthesis via incorporation into the DNA, resulting in direct inhibition of cell proliferation. However, due to the high toxicity profile of 5-AZA, other nucleoside analogs are favored as therapeutic drugs in the clinical setting [27].

Besides controlling tumor growth, epi-drugs have a strong potential to synergize with other conventional chemotherapeutic agents [29,30,31]. In the present study, all epi-drugs, except hydralazine, potentiated the anti-tumor response of gemcitabine (approximately 2-fold shift in IC_50_). This effect appeared to be primarily cytotoxic, as stated by an effect on the number of colonies rather than an effect on colony size. Co-treatment with VPA induced a synergistic effect and, importantly, it was the only epi-drug that augmented both the cytotoxic and cytostatic effect of gemcitabine on colony formation.

VPA is a well-established drug in the treatment of epilepsy, mania in bipolar disorders, and as prophylaxis of migraine headaches. Besides its known classical actions, VPA has shown potent antitumor effects in a variety of pre-clinical models by modulating multiple pathways, including cell cycle arrest, apoptosis, angiogenesis, metastasis, differentiation, and senescence [32]. Thereby, VPA inhibits HDAC enzyme activity at concentrations of 0.3–1.0 mM, which is within the therapeutic range for VPA therapy in humans [33,34]. HDAC inhibition is most probably due to binding to the catalytic center and thereby blocking substrate access. Both class I and class II HDACs are inhibited by VPA, with the highest potency for class I HDACs, in particular HDAC1 (IC_50_ for HDAC1 = 0.4 mm) [35].

Class I HDACs are preferential overexpressed in pancreatic cancer and bear poor prognostic implications [24,36,37]. For example, HDAC1 expression is positively correlated with proliferative activity, degree of tumor differentiation, and TNM staging [38]. Consistently, patients with high HDAC1 expression demonstrated a significant lower 2-year survival compared to patients with low HDAC1 expression (5% versus 32%; *p* = 0.003). Notably, high expression of both DNMT1 and HDAC1 resulted in a two-year survival of 0% com-pared to 6% in patients with only one protein highly expressed and 11% in patients that expressed both proteins at low levels (*p* < 0.001). Class I HDACs are important regulators of apoptosis and cell cycle, e.g., HDAC2 induces epigenetic silencing of the pro-apoptotic NOXA gene, whereas HDAC2 inhibition sensitized pancreatic tumor cells for tumor necrosis factor-related apoptosis-inducing ligand (TRAIL)-induced apoptosis [18,39]. Thereby, HDAC1 stimulates oxygenation of the tumor microenvironment by upregulating HIF-1α expression through HDAC1/MTA1 [40]. In addition, class I HDACs promote the epithelial-to-mesenchymal transition (EMT) of pancreatic tumor cells via the Snail/HDAC1/HDAC2 complex, which subsequently suppresses E-Cadherin expression [40].

Gemcitabine-based regimens remain essential in the treatment of patients with pancreatic cancer, but, unfortunately, resistance to gemcitabine has negatively affected overall survival rates [5]. Resistance to gemcitabine is attributable to several processes that occur in tumor cells and in the tumor microenvironment. One of these processes includes epigenetic silencing of genes that are crucial for gemcitabine’s intracellular uptake and its metabolic activation [41,42,43,44]. Targeting these genes via epi-drugs would thus potentially increase tumor cell exposure to gemcitabine and improve gemcitabine efficacy.

In the present study, we confirm the low expression of transporter genes and drug activating genes in KPC3 cells. *Slc28a1* and *Slc28a3*, both crucial drug transporter genes, could not be detected in KPC3 cells, but were evidently upregulated after epi-drug treatment. This upregulation was not observed upon treatment with the relatively weak DNMTi hydralazine [45]. Contrarily, Candelaria et al. reported a significant reversal of gemcitabine resistance by hydralazine in cervical cancer cells via upregulation of the drug transporter *SLC29A1* and the rate-limiting activating gene *dCK* [46]. However, this upregulation was independent of DNA methylation since low concentrations of hydralazine (up to 30 µM) failed to demethylate these specific promoter regions.

Although we only studied the effect of epi-drugs on *mRNA* expression, we can assume that upregulation of drug transporter genes by epi-drug treatment at least partly contributes to the synergistic effect of epi-dugs on tumor cell growth when combined with gemcitabine.

Human nucleoside transporters (hNTs) are general drug transporters that also facilitate the intracellular uptake of other nucleoside analogs, including 5-AZA [45,47,48]. Inhibition of hNT resulted in reduced cellular cytotoxicity of 5-AZA as well as its DNA demethylating activity. Other substrates for these transporters include capecitabine and 5-FU, which are also commonly given to patients with pancreatic cancer. Thereby, their expression is not only expressed in pancreatic cancer, suggesting that other cancers could also benefit from increased drug transporter expression.

The evident synergistic in vitro effect of VPA, along with its relatively safe toxicity profile, provide a strong rationale to combine VPA with gemcitabine [49]. Previous studies reported similar findings in vitro, but, importantly, in vivo data was lacking [50,51]. Hence, to the best of our knowledge, this is the first study that validated the chemosensitizing effect of the HDACi VPA in immune-competent mice bearing pancreatic tumor obtained from KPC mice [52]. This model mimics the immune phenotypic features and the aggressiveness of human pancreatic cancer. In addition, KPC tumors are predominantly resistant to gemcitabine therapy, which is consistent with pancreatic cancer patients [53,54,55].

The most frequently used gemcitabine concentration in KPC-derived mice is 50 or 100 mg/kg intraperitoneal twice-weekly, whereas the maximum tolerated dose of gemcitabine in mice is 120 mg/kg, which is similar to the equivalent dose used in patients [56]. To avoid strong cytotoxicity, we decided to use 50 mg/kg gemcitabine for in vivo combination experiments. Moreover, VPA was daily dosed at 500 mg/kg via the intraperitoneal route, as previously reported [57,58,59]. This concentration yielded efficacy in some tumor cell lines, including human pancreatic cancer cells, was shown to be safe, and, importantly, inhibited HDAC activity.

In accordance with our in vitro experiments, combination therapy resulted in significant tumor growth control and smaller tumor volumes when compared with untreated mice (74% tumor reduction; *p* = 0.003), while monotherapy with VPA or gemcitabine did not. Despite the evident upregulation of drug transporter genes in vitro, no significant changes in *mRNA* expression were observed in mice (co-)treated with VPA. Of note, expression was measured after 21 days of treatment after mice were sacrificed. The effect of VPA on *mRNA* expression was therefore potentially no longer detectable.

To gain more insight into the intratumor changes, we measured the expression of cancer-related genes using the PanCancer IO360 panel on tumors of treated and untreated mice. Interestingly, all included immune-related pathways were upregulated in mice treated with VPA plus gemcitabine compared to un-treated mice, whereas monotherapy with either gemcitabine or VPA primarily downregulated these pathways. In particular, genes involving myeloid cell activity, e.g., *Lilra5* and *Clec7a*, were significantly upregulated by combination therapy. Thereby, expression of two key transcription factors of the type I interferon (IFN) signaling pathway, *Stat2* and *Irf3*, were increased by combination therapy. Type I IFNs play a central role in immunological responses and have emerged as important key regulators in numerous anti-tumor responses [60].

Previously, combination therapy of VPA plus gemcitabine was shown to stimulate specific immune responses towards pancreatic cancer cells. For example, studies report enhanced NK cell-mediated lysis via up-regulation of cell-surface MHC class I chain-related gene A/B (MICA/B) on tumor cells [58,61]. MICA/B function as ligands for natural killer group 2 member D (NKG2D), expressed on cytotoxic innate immune cells, e.g., T cells and NK cells. It is important to address that the pro-tumoral pathways, such as angiogenesis, metastasis, and hypoxia, were not upregulated by combination therapy, but were equally downregulated as observed in the other treatment groups compared to untreated tumors. It should be emphasized, however, that we only measured *mRNA* gene expression, and thus, these results necessitate further validation at protein level.

Since animal models may deviate from humans (e.g., pharmacokinetics, pharmacodynamics, and immune reactions), our observations cannot be extrapolated to pancreatic cancer patients just like that. However, although clinical data for VPA in patients with pancreatic cancer are limited so far, the initial results look promising. In a clinical phase I/II trial, the combination therapy of VPA and S-1 for patients with pancreatobiliary tract cancers resulted in a disease control rate (partial response and stable disease) of 97.1% [62]. Importantly, adverse side effects were manageable and the degree of toxicities was comparable to those treated with S-1 alone [63,64].

## 5. Conclusions

In conclusion, we show that the class I HDACi VPA strongly potentiated the antitumor response of gemcitabine in pancreatic cancer in vitro and in vivo. Treatment with VPA primed tumor cells and increased their expression of drug-transporter genes. In addition, combining VPA treatment with gemcitabine increased the expression of immune-related genes, which strongly advocates for immune activation. Taken together, this study highlights the potential benefit of combining VPA with gemcitabine to treat pancreatic cancer patients. Additional information from phase I/II trials will provide important further data on the safety and efficacy of VPA combined with gemcitabine alone or in combination with other chemotherapeutics.

## Figures and Tables

**Figure 1 biomedicines-10-00517-f001:**
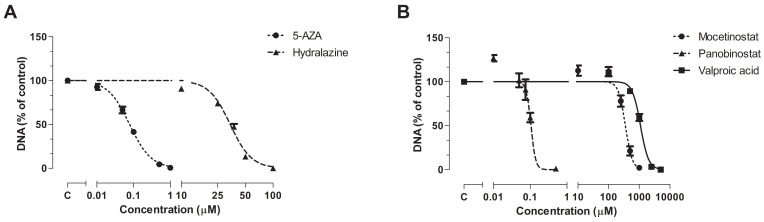
Antiproliferative effect of epi-drugs. Dose-response curves of (**A**) the DNMTis 5-AZA and hydralazine an (**B**) the HDACis mocetinostat, panobinostat, and VPA on total DNA, as a measure of cell number, in KPC3 cells after seven days of treatment. Values represent mean ± SEM of at least two independent experiments in quadruplicate and are shown as the percentage of control.

**Figure 2 biomedicines-10-00517-f002:**
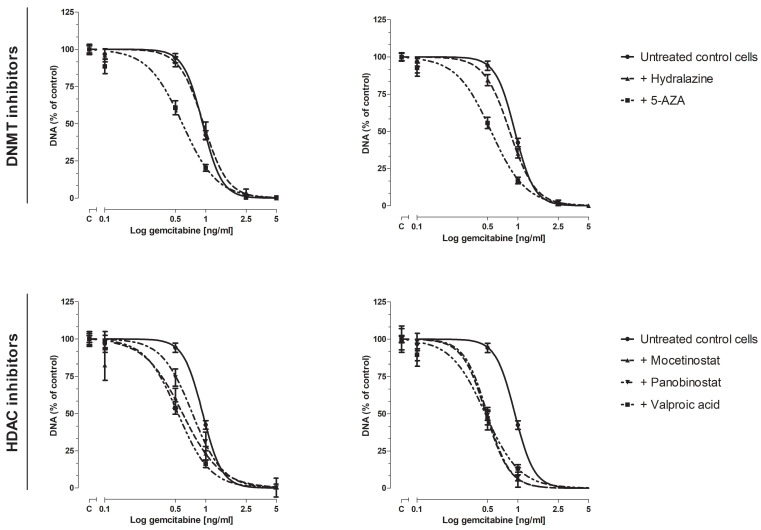
Dose-response curve of gemcitabine, with or without epi-drugs. Effect of gemcitabine, with or without epi-drugs, on cell growth in KPC3 cells after seven days of treatment. Black solid lines represent the effect of gemcitabine monotherapy in untreated control cells. Dotted lines represent the effect of gemcitabine in epi-drug treated cells (left panel IC_25_ epi-drug treated cells; right panel IC_50_ epi-drug treated cells). Data are presented as percentage of vehicle treated control. When a combination of gemcitabine with epi-drugs was examined, the control was set as the indicated epi-drug monotherapy. Values represent mean ± SEM and are shown as percentage of control.

**Figure 3 biomedicines-10-00517-f003:**
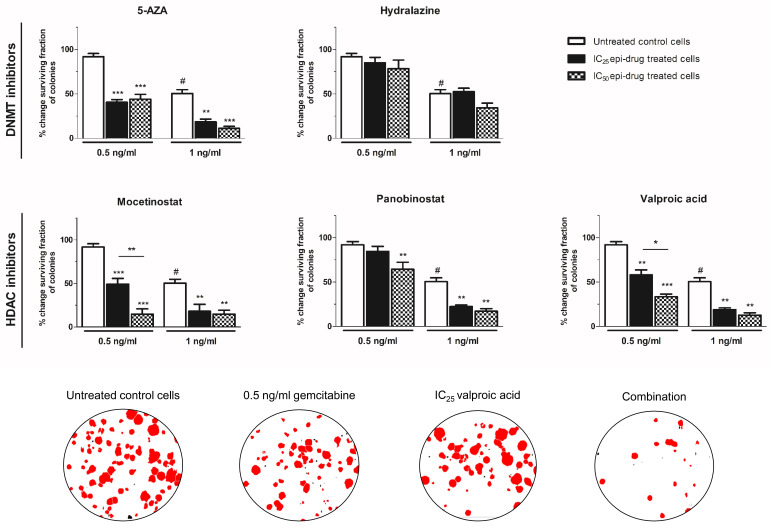
Cytotoxic effect of gemcitabine, with or without epi-drugs, on colony formation. Column bars represent the effect of seven days gemcitabine (0.5 and 1 ng/mL) in untreated control cells (white bar), IC_25_ epi-drug treated cells (black bar), and IC_50_ epi-drug treated cells (black dotted bar) on surviving fraction. Data are presented as a percentage of the vehicle-treated control. For epi-drug treated cells, the effect of the epi-drug alone was set on 100% and used as control. Photomicrographs of treatment effects on KPC3 colonies. Red stained colonies represent the measured colonies. Based on cut-off values for number and size, black stained colonies were excluded. Values represent mean ± SEM of at least two independent experiments and are shown as a percentage of control. * *p* < 0.05, ** *p* < 0.01, and *** *p* < 0.001 versus control. ^#^ *p* < 0.01 versus untreated control cells.

**Figure 4 biomedicines-10-00517-f004:**
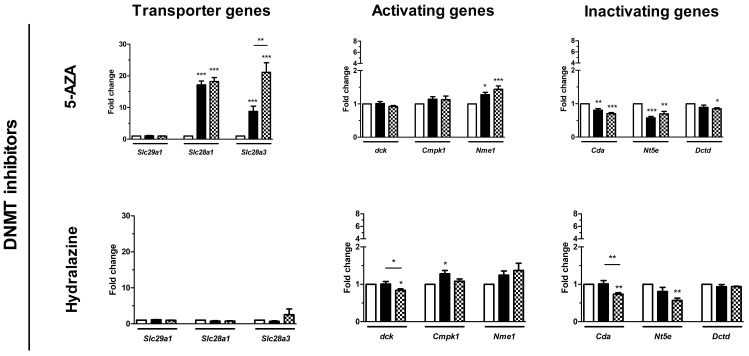
Effect of epi-drugs on *mRNA* expression of genes involved in gemcitabine metabolism. Fold change in *mRNA* expression between untreated control cells (white bar) and after seven days IC_25_ epi-drug treatment (black bar) or IC_50_ epi-drug treatment (dotted black bar). Values represent mean ± SEM of at least two independent experiments in quadruplicate and are shown relative to control (set as 1). * *p* < 0.05, ** *p* < 0.01, and *** *p* < 0.001 versus control.

**Figure 5 biomedicines-10-00517-f005:**
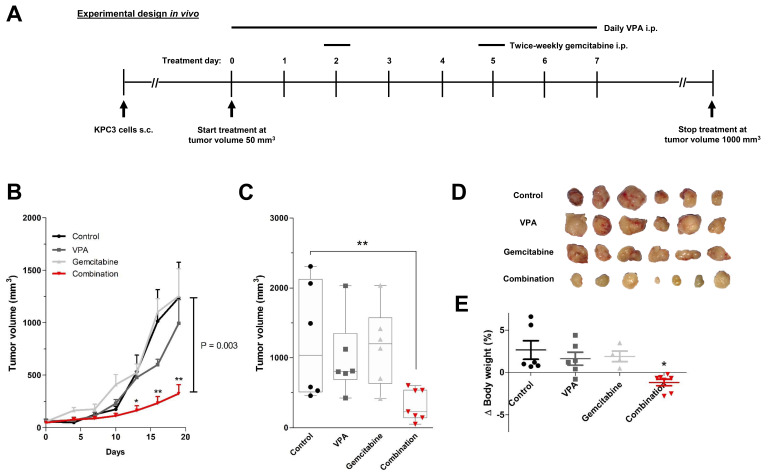
In vivo treatment effects of gemcitabine combined with VPA in immune competent mice. (**A**) Experimental design for in vivo experiments. KPC3 mouse pancreatic cancer cells (1 × 10^5^/100 µL PBS/0.1% BSA) were subcutaneously injected in C57BL/6 mice. Treatment was started when tumor volumes reached 50 mm^3^. Groups of mice received daily an intraperitoneal (i.p.) injection of VPA (500 mg/kg), two times a week (at day 2 and 5) an i.p. injection of gemcitabine (50 mg/kg), or the combination of VPA plus gemcitabine. Mice in the control group received daily an i.p. injection of 100μL distilled water. (**B**) Time course of change in tumor volume. (**C**) After 21 days of treatment, mice were sacrificed and tumor volumes were measured. (**D**) Tumor images after 21 days of treatment. (**E**) Body weight difference from start until end of treatment. Values represent mean ± SEM. * *p* < 0.05 and ** *p* < 0.01 versus control.

**Figure 6 biomedicines-10-00517-f006:**
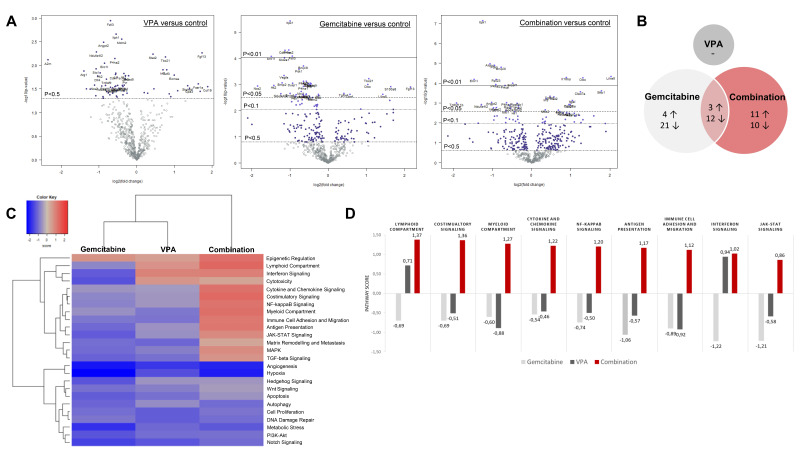
Nanostring analysis PanCancer IO 360 panel. (**A**) Volcano plot displays each gene’s −log10(*p*-value) and log2 fold change with the selected covariate. Highly statistically significant genes fall at the top of the plot above the horizontal lines, and highly differentially expressed genes fall to either side. Horizontal lines indicate various *p*-value thresholds. The 40 most statistically significant genes are labelled in the plot. (**B**) Venn diagram shows the number of up- and down-regulated genes in the three treatment arms compared to control. Overlapping areas represent the number of genes that were altered by both treatment arms. (**C**) Heatmap displays each sample’s directed global significance scores. Directed global significance statistics measure the extent to which a gene set’s genes are up- or down-regulated with the variable. Red denotes gene sets whose genes exhibit extensive overexpression with the covariate, blue denotes gene sets with extensive under expression. (**D**) Pathway scores of immune-related pathways. Increasing pathway scores corresponds to increasing expression.

**Table 1 biomedicines-10-00517-t001:** IC_50_ values of gemcitabine on cell growth in KPC3 cells with or without epi-drugs.

	IC_25_ Epi-Drug Treated Cells	IC_50_ Epi-Drug Treated Cells
**Untreated control cells**	0.9341 (0.8967–0.9731)	
**+ 5-AZA**	0.5882 (0.5418–0.6385) ***	0.5426 (0.4986–0.5904) ***
**+ Hydralazine**	0.9513 (0.8752–1.0340)	0.8298 (0.7721–0.8919)
**+ Mocetinostat**	0.5910 (0.5052–0.6913) ***	0.4834 (0.4173–0.559) ***
**+ Panobinostat**	0.7390 (0.6570–0.8313) ***	0.4972 (0.4254–0.5811) ***
**+ VPA**	**0.5320 (0.4971–0.5695) *****	**0.4758 (0.4355–0.5198) *****

IC_50_ values are presented in nanogram per milliliter (ng/mL, 95% CI). Control cells were vehicle treated. *p* values compare IC50 value of gemcitabine in untreated control cells to IC50 value of gemcitabine in epi-drugs treated cells. IC50 values depicted in bold represent the strongest decrease. *** *p* < 0.001.

## Data Availability

Not applicable.

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
