# Peer review of "The Class I HDAC Inhibitor Valproic Acid Strongly Potentiates Gemcitabine Efficacy in Pancreatic Cancer by Immune System Activation"

_biomedicines, 2022, doi:10.3390/biomedicines10030517_

Round 1
Reviewer 1 Report
In this manuscript, the authors showed that VPA strongly improved the anti-tumor efficacy of gemcitabine in pancreatic cancer. Overall, the idea and aim are reasonable, authors showed enough data about synergy effect of combination treatment, VPA and gemcitabine, in vitro & syngeneic xenograft in vivo using KPC3 pancreatic cancer cells from KPC genetic engineering mouse strain, and which seems to be very significant in almost experiments.
Meanwhile, authors emphasized this study is the first report to show new action mechanism of combination treatment (VPA & gemcitabine) by immune system activation. However, in that context, authors only suggested the possibility of an immune activation triggered by combination treatment through the genomic analysis (Nanostring analysis), but verification of this was very insufficient. If so, it is difficult to find different points from previous studies, because synergistic effects for these combination treatments in pancreatic cancer have already been reported as in Hehe Li et al, Cell Biosci. 2019; 9: 50; Yufeng Wang et al, Target Oncol. 2015;10(4):575.
Therefore, unfortunately this study seems to have limited value for novel finding to publish in Biomedicine journal.
Author Response
Response to Reviewer 1 Comments
Point 1: In this manuscript, the authors showed that VPA strongly improved the anti-tumor efficacy of gemcitabine in pancreatic cancer. Overall, the idea and aim are reasonable, authors showed enough data about synergy effect of combination treatment, VPA and gemcitabine, in vitro & syngeneic xenograft in vivo using KPC3 pancreatic cancer cells from KPC genetic engineering mouse strain, and which seems to be very significant in almost experiments.
Meanwhile, authors emphasized this study is the first report to show new action mechanism of combination treatment (VPA & gemcitabine) by immune system activation. However, in that context, authors only suggested the possibility of an immune activation triggered by combination treatment through the genomic analysis (Nanostring analysis), but verification of this was very insufficient. If so, it is difficult to find different points from previous studies, because synergistic effects for these combination treatments in pancreatic cancer have already been reported as in Hehe Li et al, Cell Biosci. 2019; 9: 50; Yufeng Wang et al, Target Oncol. 2015;10(4):575.
Therefore, unfortunately this study seems to have limited value for novel finding to publish in Biomedicine journal.
Response 1: We agree with the reviewer that the proposed immune modulation needs to be validated in future studies. We addressed this limitation now in our discussion section as an important recommendation (page 15, lines 527-528).
The reviewer refers to previous studies that also evaluated gemcitabine combined with valproate in pancreatic cancer. However, importantly, these studies lack in vivo data, while our study demonstrates a strong tumor inhibition in immune-competent mice treated with this combination regimen, which strengthens the manuscript significantly. Moreover, we used the unique KPC3 cell line, derived from the clinically relevant KPC mouse model, which carriers the major driver mutations TP53 and KRAS and mimics the immune phenotypic features, aggressiveness, and gemcitabine resistance character of human pancreatic cancer. This information and the suggested references have been added now to the discussion section of the revised manuscript (page 15, lines 527-528).
We feel, therefore, that our manuscript includes a considerable amount of novel data when compared to previously published studies.
Reviewer 2 Report
Reviewers Comments:
The authors present an interesting study in the review article entitled “The class I HDAC inhibitor valproic acid strongly potentiates 2 gemcitabine efficacy in pancreatic cancer by immune system 3 activation”. I think it would be worthwhile making major adjustments in the result section of the manuscript for acceptance in the “Biomedicines” Journal.
Major Comment:
1) Figure 3 results indicate that mocetinostat had a better cytotoxic effect than Valproate at both IC25 and IC50 concentrations in combination with Gemzar. Then what is the rationale for selecting Valproate for further experiments? Based on the results mocetinostat was the better choice for further combination studies?
2) What is the basis of the selection of 500mg/mL (Valproate) and 50ng/mL (Gemzar) concentration for mouse study? In most of the combination studies, the doses may further decrease than the monotherapy concentrations? How did the current combination concentrations adhere to the standards?
3) In the method section author mentioned the in vivo concentration as 500mg/kg (Valproate) and 50mg/kg (Gemzar). But in the result section the concentration used are 500mg/mL (Valproate) and 50ng/mL (Gemzar). How does this conversion is achieved?
4) Gemzar is administered intravenously and in the current study author selected ip as the route of administration. Please add relevant studies that have used ip administration for Gemzar treatment? If not, the study needs to be redone using the IV route.
5) Please add tumor figures for all the mice from the group and update the figure 3.4D?
6) In lines 136-137 author mentioned that “Mice were randomized in four groups and subcutaneously injected in the flank with 100.000 low passage (i.e., passage number 3) KPC3 cells in 100 µl PBS/0.1% BSA”.
What are those 100.000 cells? Please read each line thoroughly and modify the sentences.
Author Response
Response to Reviewer 2 Comments
The authors present an interesting study in the review article entitled “The class I HDAC inhibitor valproic acid strongly potentiates gemcitabine efficacy in pancreatic cancer by immune system activation”. I think it would be worthwhile making major adjustments in the result section of the manuscript for acceptance in the “Biomedicines” Journal.
We thank the reviewer for the positive general comment. Please find below a point-by-point reply to the specific comments and suggestions.
Major Comment:
Point 1: Figure 3 results indicate that mocetinostat had a better cytotoxic effect than Valproate at both IC25 and IC50 concentrations in combination with Gemzar. Then what is the rationale for selecting Valproate for further experiments? Based on the results mocetinostat was the better choice for further combination studies?
Response 1: Although mocetinostat induced a slight stronger cytotoxic effect than valproate, we chose to study valproate in vivo based on two arguments. First, valproate not only increased the cytotoxic effect of gemcitabine on colony formation, but also the cytostatic effect of gemcitabine (Supplementary Fig. 3). This effect was not observed with mocetinostat and was even significantly counteracted in the combination EC25 mocetinostat + 1 ng/ml gemcitabine. Secondly, valproate is a well-known drug, from which the pharmacokinetic and pharmacodynamics are well understood. Thereby, valproate has a relative safe toxicity profile compared to other epi-drugs.
Point 2: What is the basis of the selection of 500mg/mL (Valproate) and 50ng/mL (Gemzar) concentration for mouse study? In most of the combination studies, the doses may further decrease than the monotherapy concentrations? How did the current combination concentrations adhere to the standards?
Response 2: We have added a justification, including relevant references, for the chosen drug concentrations to the discussion section of the revised version of the manuscript (discussion section, page 14, lines 478-488).
“In addition, KPC tumors are predominantly resistant to gemcitabine therapy, which is consistent with pancreatic cancer patients (53-55). The most frequently used gemcitabine concentration in KPC-derived mice is 50 or 100 mg/kg intraperitoneal twice-weekly, whereas the maximum tolerated dose of gemcitabine in mice is 120 mg/kg, which is similar to the equivalent dose used in patients (56). To avoid strong cytotoxicity, we decided to use 50 mg/kg gemcitabine for in vivo combination experiments. Moreover, VPA was daily dosed at 500 mg/kg, via intraperitoneal route, as previously reported (57-59). This concentration yielded efficacy in some tumor cell lines, including human pancreatic cancer cells, was shown to be safe, and, importantly, inhibited HDAC activity.”
Point 3: In the method section author mentioned the in vivo concentration as 500mg/kg (Valproate) and 50mg/kg (Gemzar). But in the result section the concentration used are 500mg/mL (Valproate) and 50ng/mL (Gemzar). How does this conversion is achieved?
Response 3: We thank the reviewer for his alertness. The correct concentration is 500 mg/kg VPA and 50 mg/kg gemcitabine. We have changed this in the revised version of the manuscript (results section, page 9). We apologize for this omission.
Point 4: Gemzar is administered intravenously and in the current study author selected ip as the route of administration. Please add relevant studies that have used ip administration for Gemzar treatment? If not, the study needs to be redone using the IV route.
Response 4: Although gemcitabine is administered intravenously in patients, in vivo studies use intraperitoneal administration. We now added relevant references in the revised manuscript (discussion section, page 14, line 478-488).
Point 5: Please add tumor figures for all the mice from the group and update the figure 3.4D?
Response 5: We thank the reviewer for this suggestion and updated the figure (results section, page 10)
Point 6: In lines 136-137 author mentioned that “Mice were randomized in four groups and subcutaneously injected in the flank with 100.000 low passage (i.e., passage number 3) KPC3 cells in 100 µl PBS/0.1% BSA”.
What are those 100.000 cells? Please read each line thoroughly and modify the sentences.
Response 6: We understand the confusion and modified the sentence to: “Mice were randomized in four groups and subcutaneously injected in the flank with 100.000 KPC3 cells (passage number 3) in 100 µl PBS/0.1% BSA” (method section, page 4).
Round 2
Reviewer 1 Report
The authors improved the manuscript by rephrasing the following content in response to the reviewers' comments. The synergistic effect of VPA in vivo level was first verified using the KPC mouse-derived cell transplantation model, which is valid for confirming tumor immunity in pancreatic cancer. In addition, the requirement for protein-level verification at least was openly stated. By incorporating them, the distinction and significance of this paper were clarified.